# Challenges faced by human resources for health in Morocco: A scoping review

Wafaa Al Hassani[1], Youness El Achhab[2,3]*, Chakib Nejjari[3,4]

**1** Euromed of Nursing Sciences and Health Technics, Euromed University of Fez, Fez, Morocco, **2** CRMEF Fez-Meknes, Fez, Morocco, **3** Department of Epidemiology, Clinical Research and Community Health, Faculty of Medicine, Dental Medicine and Pharmacy of Fez, Fez, Morocco, **4** Euromed Research Center, Euromed University of Fez, Fez, Morocco

* youness_elachhab@yahoo.fr

**Data Availability Statement:** All relevant data are within the paper and its Supporting Information files.

**Funding:** The author(s) received no specific funding for this work.

## Abstract

### Background

Human resources for health (HRH) play a pivotal role in effective health system operation, yet various impediments challenge sustainable development. This scoping review aimed to explore these challenges and potential solutions in aligning the health workforce to meet the evolving healthcare needs of the Moroccan population.

### Methods

We conducted a scoping review searching PubMed, Science Direct, Cairn and Google Scholar for relevant articles published between 2014 and 2023. Additionally, non-peer-reviewed literature sourced from Ministry of Health consultations and allied websites was included.

### Results

Among the nineteen studies meeting our inclusion criteria, the majority were cross-sectional and predominantly focused on challenges faced by nurses. While some papers delineated multiple HRH challenges (5/19), the rest addressed specific challenges. The identified challenges span organizational and personal levels. Organizationally, the focus was on training, lifelong learning, continuing education, health coverage and shortages, and job satisfaction. At a personal level, HRH in the public health sector encountered challenges such as burnout, stress, and broader occupational health concerns.

### Conclusions

The reviewed publications underscored a spectrum of challenges necessitating robust policy interventions. Despite promising developments in the Moroccan healthcare system, addressing the unequal urban-rural HRH distribution, augmenting funding, and enhancing HRH quality of life stand as pivotal imperatives.

**Competing interests:** The authors have declared that no competing interests exist.

## Introduction

Human resources for health (HRH) serve as a linchpin for robust health systems worldwide, yet multiple challenges impede their sustainable development [1–5]. In 2016, the World Health Assembly introduced the Global Strategy on Human Resources for Health: Workforce 2030, aimed at expanding the availability, accessibility, acceptability, coverage, and quality of HRH. This strategy stands as a critical step toward achieving universal health coverage and the Sustainable Development Goals [5].

Recent progress in research has delved into various facets of HRH, examining governance, training, performance, recruitment strategies, motivation, and distribution [6–11]. Systematic reviews play a pivotal role in visualizing this evidence for policymakers. They consolidate research, shedding light on key issues affecting the health workforce within a robust health system [12, 13]. Such reviews not only highlight areas with existing evidence but also pinpoint those necessitating more attention. Ultimately, fostering a synergistic interaction between researchers and policymakers through these systematic reviews facilitates the translation process within HRH policymaking. This collaborative approach aims to bridge the gap between research insights and policy implementation.

Morocco's healthcare system is integral to the well-being of its populace, yet it grapples with a myriad of challenges, as do many nations [14]. Despite notable strides in recent years, this system confronts ongoing hurdles that demand readiness from healthcare professionals. The HRH sector stands as a critical pillar within this system, encompassing healthcare professionals, support staff, and administrators. The Joint Learning Initiative on HRH emphasizes the significance of investing in HRH as a means to explicitly elevate national income per capita and alleviate absolute poverty [15]. Moreover, participants from diverse backgrounds at the fifth Global Forum on HRH in 2023 underscored the imperative of safeguarding, protecting, and investing in the health and care workforce [16].

This paper delineates the outcomes of an initiative aimed at investigating challenges and potential solutions to optimize the health workforce. Through an extensive literature review, the objective was to align the health workforce with the evolving healthcare requirements of the Moroccan population.

## Methods

### Data sources and search strategy

Adhering to the framework outlined by Arksey and O'Malley [17], as updated by Levac *et al.* [18], we undertook a scoping review of the literature. Our aim was to delve into the challenges and potential solutions concerning the optimization of the health workforce in Morocco. Initially, the research question was formulated through consultations with an expert panel comprising members from the health ministry, an EMRO representative, and a researcher equipped with formal information management training. To establish relevant search criteria, keywords were identified based on these research inquiries and insights provided by the expert panel (S1 Appendix). Furthermore, the summary and reporting of this scoping review were done using the checklist for the Preferred Reporting Items for Systematic Reviews and Meta-Analyses extension for Scoping Reviews (PRISMA-ScR) [19] (S2 Appendix). Ethical approval was not deemed necessary for this review since the findings were derived from existing and publicly available literature.

We conducted searches on electronic databases including PubMed, ScienceDirect, and Google Scholar up to November 2, 2023, to identify pertinent articles published between 2014 and 2023 (S1 Appendix). Our search encompassed English and French articles, reviews,

abstracts, case reports, letters to the editor, and other academic reports. Additionally, to augment our search, we performed a backward citation analysis of relevant papers and incorporated recommendations provided by our expert panel.

The following key words were identified and used in various combinations with Boolean operators (and, or): health human resources, health care delivery, health workforce, health professionals, healthcare providers, personnel, nurses, doctors, shortage, challenges, barriers, opportunities, satisfaction, stress, burnout, occupational health, Morocco.

A three-stage method was adopted to select publications for review. In the first, the title alone was examined, followed by looking at the abstract, and then examining the whole publication. The search and review results are shown in Fig 1. Articles were excluded if they did not refer to a HRH, or if did not address a challenge context. All documents relevant to the COVID-19 pandemic were excluded from this scoping review. If articles were representative of the inclusion criteria, the documents went through two full-text independent reviews by the two authors.

## Quality assessment

To assess the literature quality, parts of the method presented by Hölbl *et al.* [20] were used and modified as appropriate. The reviewers checked three considerations about relevance to HRH and challenges and appropriateness of research objectives (Q1- Is the HRH challenge described? Q2- Are the research objectives clearly outlined? Q3- Are the proposed solutions feasible?).

## Data extraction

The research team extracted data for each selected article according to the following domains: author(s) name, year of publication, study design and sample size, type of HRH, type of

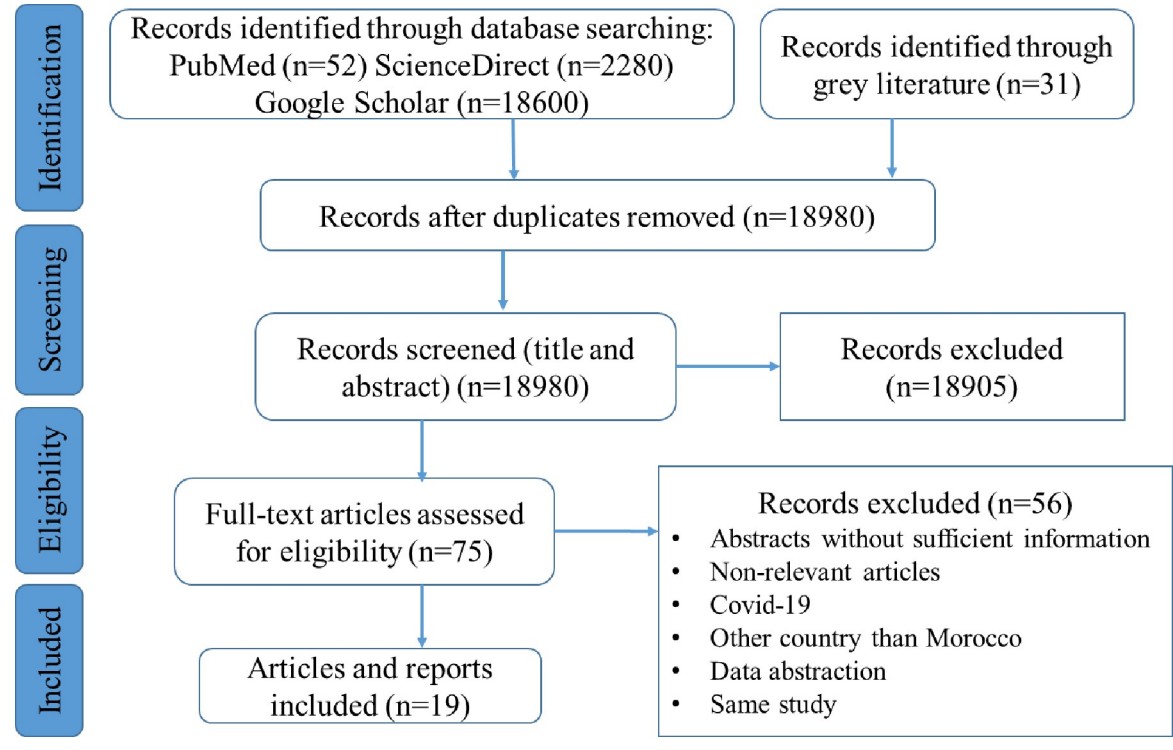

**Fig 1. PRISMA flowchart for the study selection process.**

document (peer-reviewed or gray), setting, HRH challenge, main findings, and proposed solutions or recommendations (S3 Appendix). Data analysis was completed in Microsoft Excel.

## Results

The selection process is shown in Fig 1. Initially, 18,980 titles were identified through database searches and other sources. Following the screening of titles and abstracts, 75 studies underwent full-text review. Ultimately, 19 studies met the criteria for inclusion in this scoping review. These studies spanned a 10-year range, with the earliest publication dating back to 2014. Interest in challenges encountered by healthcare providers notably surged between 2017 and 2023.

The data extracted from the included studies are presented in Table 1. The majority of publications (15 out of 19) underwent peer review. Among the included studies, half were

Table 1. Characteristics of studies selected for inclusion.

| Source | Type of study | Sample size | Participants | Publication | Setting |
|---|---|---|---|---|---|
| Elkachradi et al. (2023) [27] | Qualitative | 48 interviews | Nurses | Peer-reviewed | 3 Public hospitals |
| Fattahi et al. (2023) [36] | Cross-sectional | n = 2174 | Medical, Paramedical, Admin and Assistants | Peer-reviewed | Public health structures in Drâa-Tafilalet region |
| Dankoly et al. (2023) [21] | Qualitative | 8 focus groups | Doctors and nurses caring for diabetes patients | Peer-reviewed | 8 primary healthcare centers urban and rural regions in Oujda |
| Mansoury et al. (2023) [37] | Cross-sectional | n = 191 | Physicians and nurses | Peer-reviewed | Marrakech University Hospital |
| El Mouaddib et al. (2023) [30] | Cross-sectional | n = 346 | Nurses, GPs and specialists | Peer-reviewed | Marrakech public health centers |
| Koubri et al. (2022) [31] | Mixed study | n = 272; 18 interviews | Primary healthcare providers and officials | Peer-reviewed | Primary healthcare facilities in the Rabat Salé Kénitra region |
| Hababa et al. (2021) [24] | Case study | | Medical, Paramedical, Admin and Assistants | Peer-reviewed | Marrakech-Safi regional directorate of health |
| Belrhiti et al. (2020) [6] | Multiple embedded case study design | 68 interviews and 15 FG | Officials, doctors, nurses and administrators | Peer-reviewed | 4 Public hospitals |
| El Azizi et al. (2019) [33] | Cross-sectional | n = 80 | Nurses | Peer-reviewed | Hospital Hassan II of Settat |
| Barich et al. (2019) [25] | Curriculum analysis | | Nurses | Peer-reviewed | ISPITS of Rabat |
| Gouifrane et al. (2018) [26] | Cross-sectional | n = 129 | Nurses | Peer-reviewed | Ibn-Rochd Hospital Casablanca |
| Ministry of health (2018) [32] | Cross-sectional | n = 2122 | All HRH | Report | National |
| Chtibi et al. (2018) [38] | Cross-sectional study | n = 100 | Physicians and nurses | Peer-reviewed | Ibn Sina Hospital in Rabat |
| Laraqui et al. (2017) [35] | Cross-sectional | n = 1951 | Doctors and paramedics | Peer-reviewed | 3 Public hospitals |
| Ministry of health (2017) [28] | Mixed survey | N/A | All HRH | Report | National health institutions |
| Ministry of health (2017) [23] | Qualitative | N/A | All HRH | Report | National health institutions |
| Giurgiu et al. (2015) [34] | Cross-sectional study | n = 2863 | All HRH | Peer-reviewed | Public hospitals from north, center and south. |
| Zouag et al. (2015) [29] | Simulation | N/A | Doctors | Report | National |
| El Kirate and Filali (2014) [22] | Multiple embedded case study | 22 interviews | Mental health practitioners, senior managers and psychiatry teachers | Peer-reviewed | Health institutions in Rabat |

designed as quantitative cross-sectional (9 out of 19), while four studies adopted a mixed-method approach. Out of the studies reviewed, four publications did not specify their geographical focus. In contrast, eleven studies clearly identified the regional scope of their investigation, and the remaining four explored challenges at the national level. While the majority of studies (14/19) reported challenges faced by medical professionals, paramedics, and assistants, four studies specifically addressed challenges within one target population, namely nurses and one was focused exclusively for doctors.

Table 2 outlines the challenges encountered by HRH along with proposed solutions derived from the included studies. While five papers highlighted multiple challenges faced by HRH, the remaining addressed specific challenges. Notably, two papers delineated barriers within distinct departments: the first focused on managing Type 2 Diabetes (T2D) [21], while the second discussed challenges within the mental health system [22].

Across the included studies, a spectrum of challenges confronted by HRH emerged within both regional and national healthcare sectors. These challenges were categorized into two levels:

## Organizational level

The most frequently reported HRH challenge in the Moroccan context was training, lifelong learning, and continuing education, highlighted in five papers [24–28]. The Ministry of Health study [23] revealed that continuous training remains significantly limited due to an unstructured system and constrained by insufficient funding. The recommendation from this study emphasizes the necessity of implementing an efficient and coherent organization for continuing education while enhancing the pursuit of quality and attractiveness to strengthen its effectiveness.

Responding to the Ministry of Health's call for continuous training (outlined in the national strategy of continuous training 2019–2021), the regional directorate of health in Marrakech-Safi devised its regional plan in 2019 [24]. The study demonstrated that participants acquired skills and effectively applied them in practical settings. Among the remaining three papers, all centered on nurses' training. One study highlighted the need for a secondary reform in the training curriculum [25], while another focused on the inadequacy of motivation for continuous training [26]. Regarding lifelong learning, Elkachradi *et al.* reported that all healthcare professionals interviewed recognized the importance of training in delivering quality and safe care [27].

Two studies identified health coverage and shortages as key challenges. The first study, conducted by the Ministry of Health [28], highlighted an inadequate stock of healthcare workers that does not align with health service needs. Additionally, it noted an imbalanced distribution of healthcare personnel concerning the geographical spread of the population and its epidemiological profile. The second study, by Zouag *et al.* [29], focused on forecasting shortages through simulation for the period 2010–2030. It projected a significant deficit in medical doctors in Morocco in the upcoming years. This shortage could further escalate due to current emigration trends and retirement rates.

Three studies shed light on job satisfaction as a noteworthy challenge, characterized by a complex interplay of factors that might vary based on the specific group under study. Overall, dissatisfaction among healthcare providers is prevalent, encompassing various facets of their roles [30–32].

Belrhiti *et al.*'s study [6] highlights that enhancing public service motivation, organizational commitment, and extra-role behaviors could be achieved by amplifying perceived supervisor and organizational support, along with fulfilling staff's basic psychological needs. El Azizi and

**Table 2. Distribution of the challenges reported in the includes studies.**

| Source | Participants | Challenges | Main results | Proposed solutions |
|---|---|---|---|---|
| Elkachradi et al. (2023) [27] | Nurses | Lifelong learning | Only a quarter of the interviewed caregivers comprehend "lifelong learning". However, all of them acknowledge "continuous learning" particularly amidst prolonged uncertainty, within Moroccan public hospitals. | raise awareness about structuring lifelong professional training while integrating a knowledge management approach |
| Fattahi et al. (2023) [36] | Medical, Paramedical, Admin and Assistants | Burnout syndrome | prevalence rate of burnout (57.7%). Highly stressful working conditions. | Improve working conditions and financial motivation of public health professionals |
| Dankoly et al. (2023) [21] | Doctors and nurses caring for diabetes patients | T2D management barriers | Excessive workload, poor reimbursement policy, shortage, poor working environment, limited referral, gap in the knowledge of general practitioner. | Staff recruitment; continuous professional development; internships. |
| Mansoury et al. (2023) [37] | Physicians and nurses | Burnout syndrome | High degree of emotional exhaustion (60%). High degree of depersonalization. Low degree of personal accomplishment (73%). | Improve working conditions by reduction workload. Increase motivation and lifelong professional training. Increase manpower. |
| El Mouaddib et al. (2023) [30] | Nurses, General practitioners (GPs) and specialists | Job satisfaction | Presence of dissatisfaction among HCPs regarding various aspects of their jobs (relationship with the immediate supervisor, relationship with colleagues, administrative support, level of responsibility, and working hours). | Clarify the tasks of all professional categories, grant them more autonomy, establish career paths for the promotion and professional development, improve working conditions, and promote incentives. |
| Koubri et al. (2022) [31] | Primary healthcare providers and officials | Satisfaction with organization of health services | Presence of dissatisfaction among HCPs regarding organization of health services. 75% of the providers expressed their full agreement in reorienting PHC facilities towards the person-centered approach. | Provision of sufficient and qualified human resources, to be able to develop person-centered health services. Implementing family health as a new paradigm for the organization of health services. |
| Hababa et al. (2021) [24] | Medical, Paramedical, Admin and Assistants | Continuing education | Success of the regional continuing education plan for professionals in the Marrakech-Safi region 2019–2021. The participants acquired skills and succeeded in transferring what they acquired in the field. | Developing HRH capacities and improving the performance of the healthcare system. |
| Belrhiti et al. (2020) [6] | Officials, doctors, nurses and administrators | Public service motivation | Complex leadership styles can increase public service motivation, organizational commitment and extra role behaviors by increasing perceived supervisor and organizational support and satisfying staff basic psychological needs. | In hospitals, the archetype of complex professional bureaucracies, leaders need to be able to balance between different leadership styles according to the staff's profile, the nature of tasks and the organizational culture. |
| El Azizi et al. (2019) [33] | Nurses | Evidence-based nursing practice | Organizational barriers, barriers related to nurses' personal characteristics, research barriers, and institutional barriers. | Increase motivation and lifelong professional training. Ensuring access to information sources (scientific databases). |
| Barich et al. (2019) [25] | Nurses | Nursing training | Theoretical courses predominate for all sectors. Objective-based approach instead of skills-based approach. Insufficient time for training. | Need of a second reform referring to the required standards and the specificities of training in nursing and health techniques. |
| Gouifrane et al. (2018) [26] | Nurses | Motivation for continuous training | Level of motivation for continuous training (64%). Extrinsic motivation exceeds intrinsic motivation. | None |
| Ministry of health (2018) [32] | All HRH | Satisfaction, engagement and motivation | Lack of satisfaction (64%). Excess of workload (75%). Stress (75%). 68% unsatisfied from remuneration. 78% satisfied from motivation of their leaderships. Need of continuous training (64%). | Optimal and appropriate sizing of resources reduces overload and improves the working climate. Ad hoc support structures and innovative solutions contribute to coping with stress. |
| Chtibi et al. (2018) [38] | Physicians and nurses | Stress and Burnout | Emotional exhaustion (42%), depersonalization (49%), low professional achievement (67%), and psychological distress (54%). The stress test reveals that 88% of subjects have a low or moderate level of stress resistance. | None |

(*Continued*)

**Table 2.** (Continued)

| Source | Participants | Challenges | Main results | Proposed solutions |
|---|---|---|---|---|
| Laraqui et al. (2017) [35] | Doctors, nurses and Assistants | Well-being and occupational perception | Medication use consisted of analgesics (28.1%) and psychotropic (11.6%). Bad general health perception (38.6%). Pain and headache were the most common symptoms. 53.9% experienced stress at work. Difficult work requirements (78.3%). | Assessing the well-being, working conditions and perceived occupational hazards of healthcare staff should be a priority for occupational health departments in Moroccan hospitals. |
| Ministry of health (2017) [28] | All HRH | Continuing training | Diagnosis: Continuing training is very limited, based on an unstructured system and hampered by the lack of funding. | Deploy an efficient and coherent organization for continuing education. strengthen skills to support the Ministry of Health's 2017–2021 sector strategy |
| Ministry of health (2017) [23] | All HRH | Health coverage | Insufficient stock of healthcare workers. Scarce information on dynamics of the healthcare labor market. Maldistribution of healthcare personnel. Insufficient training to guarantee the quality of services. | Increase the availability of HRH according to the service needs. Achieve universal coverage especially in regions with deficits. Design and implement strategies to improve the technical quality of HRH. |
| Giurgiu et al. (2015) [34] | All HRH | Occupational risk perception | Shortage in doctors and nurse staff. Aggression, workload, stress, and musculoskeletal disorders. HCWs took more hypnotics, sedatives and analgesics (40%). | Urgent need of implementing a risk prevention plan and even a hospital reform. |
| Zouag et al. (2015) [29] | Doctors | Personnel Shortage | The baseline scenario shows already important deficits in medical doctors in Morocco for the coming years. This can only increase under the current emigration flows and the retirement rates. | The cooperative frameworks with other countries and mainly EU could be an important source for satisfying both the needs of the EU and those of Morocco. |
| El Kirate & Filali (2014) [22] | Mental health practitioners, senior managers and psychiatry teachers | Challenges faced by the mental health system | Lack of communication, collaboration and updated knowledge especially concerning diagnosis, psychotropic drug prescriptions and addiction medicine. Need of specific training in mental health specialists, such as geriatric psychiatry and pediatric psychiatry. | Better training in these areas would contribute to the success of managed care strategies in primary healthcare facilities. |

El Goundali's research [33] focuses on delineating barriers to the integration of evidence-based practices in nursing. It identifies four categories of barriers that could potentially underlie reluctance towards evidence-based practice.

## Personal level

Two studies focused on occupational health highlighted that healthcare workers in Moroccan hospitals tend to use more hypnotics, sedatives, and analgesics [34, 35]. Urgent recommendations emerge from these findings, emphasizing the crucial need to assess the well-being, working conditions, and perceived occupational hazards among healthcare staff.

Additionally, two studies underscored the heavy workload experienced within Morocco's public hospitals, contributing to stress and burnout among healthcare professionals [36, 37]. Another study aimed to scrutinize the relationship between resistance status, burnout, and levels of psychological distress [38], revealing that 88% of subjects demonstrated a low or moderate level of stress resistance. Most studies included in this analysis corroborate this stressful scenario and advocate for improving working conditions, as well as promoting incentives and compensation.

## Discussion

This scoping review delved into the challenges confronting HRH within the workforce, as identified in both scientific and grey literature. The study's robustness lies in its rigorous inclusion criteria and the guidance provided by a panel of experts in shaping the approach to the

subject. Notably, nurses emerged as more prominently featured among all professions within the reviewed literature. However, several selected publications encompassed all health professions, primarily due to the comprehensive nature of challenges examined, especially those concerning national health policies.

While a majority of the included studies relied on quantitative methods, the utilization of mixed methods presents an opportunity for triangulating findings. This approach can enrich the contextual understanding of challenges and aid in crafting specific recommendations. Within the body of knowledge on HRH challenges in this scoping review, empirical research takes precedence. Our findings underscore the tendency of empirical studies to concentrate on specific challenges rather than providing a holistic overview. Indeed, the results gleaned from these publications emphasize that challenges are interconnected and often overlap.

Our findings underscore that the most extensively studied challenges since 2014 predominantly align with organizational and personal spheres. Organizational challenges encompass four primary areas: training, lifelong learning, and continuing education; health coverage and shortages; and job satisfaction. Meanwhile, at the personal level, HRH in the public health sector face challenges including burnout, stress, and broader issues related to occupational health. It's noteworthy that this scenario is not unique to Moroccan HRH but is mirrored in numerous countries worldwide [39–42].

In our digital society, health systems encounter significant challenges, prompting the need to reform models rooted in the industrial age [43]. Responding to a dynamic and complex global environment, the World Health Organization has introduced the 'Human Resources for Health 2030 strategy' as a new global paradigm [44]. Central to the realization of this vision is the development of a novel regulatory model. To modernize its regulatory framework, the Moroccan Ministry of Health has devised new strategies in collaboration with the WHO [6, 14, 45].

The pressing demand to train more healthcare providers to address an estimated shortfall of 18 million, primarily in low- and middle-income countries, presents a critical challenge for health policymakers [5]. One promising strategy to augment the number of medical professionals involves reforming admission and training practices in medical education [46–48]. The focus on training and the future prospects of the health workforce for the 2010–2030 period, as studied by Zouag et al. [29], emphasizes the necessity for enhanced cooperation in healthcare, particularly emphasizing medical education and research. Notably, a study on development assistance for HRH [49] highlighted a significant portion of support directed toward HRH-related training activities between 2016 and 2019.

Following the structuring of continuing training by the Moroccan Ministry of Health [23], local health directorates have devised their strategies. The study conducted by Hababa et al. [24] showcased the effectiveness of a training program executed in 2019 in bolstering HRH skills. Moreover, healthcare professionals view lifelong learning as an ongoing and indispensable process for their work [27, 50]. Furthermore, a well-trained HRH is an indispensable element for the success of any healthcare system [51].

This new era is marked by significant transitions, including epidemiologic shifts and a redistribution of the disability burden [4]. These changes significantly impact healthcare systems, the roles of HRH, and educational policies [52, 53]. Such transitions further exacerbate the prevalent global shortage and uneven distribution of HRH [1, 4].

In Morocco, the Ministry of Health conducted a national study on health coverage [28], revealing an inadequate supply of healthcare workers that doesn't align with health service needs. Additionally, there's an imbalanced distribution of healthcare personnel concerning the geographical spread of the population and its health profile. The shortage is compounded by HRH emigration. A recent WHO report indicated that the escalating demand for healthcare

providers in developed countries might heighten vulnerabilities in nations already grappling with low health workforce densities [54]. In response, a study in Morocco recommends enhancing working conditions, improving training quality, and revisiting HRH salaries to mitigate medical student migration [55].

The satisfaction of health professionals with their jobs significantly influences HRH performance and the quality of healthcare services [56, 57]. Within this scoping review, dissatisfaction among health professionals across various job aspects is evident [30–32]. Globally, particularly in developing nations, several factors contribute to this decreased job satisfaction among health professionals. Management and leadership, working conditions, staff relations and patient care, responsibility, and workload emerge as key factors linked to job satisfaction [57–59].

Introducing effective interventions has shown promise in enhancing nurses' job satisfaction and curbing employee turnover [56, 57, 60]. Additionally, factors such as job security, recognition, respect, financial independence, and competitive salaries are reported in the literature to heighten HRH motivation [61]. Notably, in a national survey of US physicians, intrinsic motivational factors were associated with physicians' career enjoyment, life satisfaction, and commitment to clinical practice [62].

According to several studies [60, 63–65], job satisfaction has been found to correlate with stress levels and the incidence of burnout symptoms within work environments. Stress represents a universal element in the job of healthcare professionals [66]. Chronic occupational stress, heavy workloads, and imbalances between job demands and resources are primary contributors to HRH burnout [67, 68].

A meta-analysis of 47 studies revealed that physician burnout increased the risk of medical errors and led to diminished overall care quality and patient satisfaction [69]. However, findings from a review and meta-analysis underscored the effectiveness of cognitive, behavioral, and mindfulness-based approaches in reducing stress levels and burnout among HRH [70]. In our context, studies advocate for enhancing working conditions and providing better financial motivation for public health professionals to mitigate stress and burnout [36, 37].

## Limitations

This scoping review bears certain limitations. Firstly, the internal and external validity of the findings might hinder their generalizability to the entire HRH spectrum. The risk of selection bias is a possibility, as only two researchers conducted and agreed upon the database selection. Moreover, the majority of studies included in this review primarily originated from major cities or prominent regions, potentially limiting the diversity and representation of HRH experiences across various settings. Secondly, the predominance of cross-sectional studies focused predominantly on nurses within HRH, along with disparities in sample sizes and variations in the measurement tools used to assess challenges, could impact the broader applicability of the findings.

## Conclusion

This scoping review gathered pertinent documents to offer insights into the prevailing challenges within HRH in Morocco. The identified publications delineated a diverse array of challenges that necessitate robust policy interventions. Despite recent strides in the Moroccan healthcare system, tackling the disparities in HRH distribution between urban and rural areas, augmenting funding, and enhancing the quality of life for HRH remain pivotal. This requires concerted efforts towards strategic planning for staff training, coupled with the implementation of improved HR management practices and enhanced motivation within regional public health institutions.

## Supporting information

**S1 Appendix. Search strategy.**
(DOCX)

**S2 Appendix. PRISMA checklist.**
(DOCX)

**S3 Appendix. Data extraction sheet.**
(DOCX)

## Author Contributions

**Conceptualization:** Wafaa Al Hassani, Youness El Achhab, Chakib Nejjari.

**Data curation:** Wafaa Al Hassani, Youness El Achhab.

**Formal analysis:** Wafaa Al Hassani, Youness El Achhab.

**Methodology:** Wafaa Al Hassani, Youness El Achhab, Chakib Nejjari.

**Supervision:** Chakib Nejjari.

**Validation:** Chakib Nejjari.

**Writing – original draft:** Wafaa Al Hassani.

**Writing – review & editing:** Youness El Achhab, Chakib Nejjari.

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
