## [Decision Letter · Decision Letter 0]

4 Feb 2024

PONE-D-23-42584Challenges faced by Human Resources for Health in Morocco: a scoping reviewPLOS ONE

Dear Dr. youness,

Thank you for submitting your manuscript to PLOS ONE. After careful consideration, we feel that it has merit but does not fully meet PLOS ONE’s publication criteria as it currently stands. Therefore, we invite you to submit a revised version of the manuscript that addresses the points raised during the review process.

**Dear Authors**Please respond to the reviewers' comments as soon as possible. The answers should be clear and point-by-point the text.

We look forward to receiving your revised manuscript.

Kind regards,

Morteza Arab-Zozani, Ph. D.

Academic Editor

PLOS ONE

Reviewers' comments:

Reviewer's Responses to Questions

**Comments to the Author**

1. Is the manuscript technically sound, and do the data support the conclusions?

Reviewer #1: Yes

Reviewer #2: Yes

Reviewer #3: No

2. Has the statistical analysis been performed appropriately and rigorously? 

Reviewer #1: Yes

Reviewer #2: N/A

Reviewer #3: No

3. Have the authors made all data underlying the findings in their manuscript fully available?

Reviewer #1: Yes

Reviewer #2: Yes

Reviewer #3: No

4. Is the manuscript presented in an intelligible fashion and written in standard English?

Reviewer #1: Yes

Reviewer #2: Yes

Reviewer #3: No

5. Review Comments to the Author

Reviewer #1: This is a well written manuscript. It highlights the challenges faced by human resources for health in the context of Morocco. This manuscript highlights the lack of policies related to the HRH. This gives concise information about the problem among HRH on a personal level as well as institutional level. This will be helpful for concerned authorities while making new policies for health professionals. This manuscript also compares those problems with other developing nations as well as the developed nation.

Reviewer #2: It followed rigorous methodology that has yielded into good study findings and conclusion. Though, study data is available, the data extraction sheet for selected papers could also be provided. Authors can also elaborate about the consultations done to develop the research question.

Reviewer #3: The effort committed into the manuscript is appreciated. However, authors may wish to note that the manuscript has not clearly answered the research question. In addition, the flow of the manuscript needs to be carefully worked on to appeal to readers. In the table, separate column should be used, each for type of study and the sample size. It seems the manuscript has been published on a blog (accessed -30-1-24) https://sciety.org/articles/activity/10.1101/2023.12.21.23300411

6. PLOS authors have the option to publish the peer review history of their article (what does this mean?). If published, this will include your full peer review and any attached files.

Reviewer #1: **Yes: **Rajeev Mishra

Reviewer #2: **Yes: **Sanjeev Kumar

Reviewer #3: No

---

## [Author Response · Author response to Decision Letter 0]

17 Feb 2024

Dear Editor,

Thank you for giving us the opportunity to submit a revised draft of the manuscript for publication in the PLOS One journal. We appreciate the time and effort that you and the reviewers dedicated to providing feedback on our manuscript and are grateful for the insightful comments on and valuable improvements to our paper. We have incorporated most of the suggestions made by the reviewers. Those changes are highlighted within the manuscript. We hope the manuscript after careful revisions meet your high standards.

Please see below, in blue, for a point-by-point response to the reviewers’ comments and concerns.

Sincerely,

Youness EL ACHHAB

---

## [Decision Letter · Decision Letter 1]

4 Mar 2024

Challenges faced by Human Resources for Health in Morocco: a scoping review

PONE-D-23-42584R1

Dear Dr. youness,

We’re pleased to inform you that your manuscript has been judged scientifically suitable for publication and will be formally accepted for publication once it meets all outstanding technical requirements.

Kind regards,

Morteza Arab-Zozani, Ph. D.

Academic Editor

PLOS ONE

Additional Editor Comments (optional):

Reviewers' comments:

Reviewer's Responses to Questions

**Comments to the Author**

1. If the authors have adequately addressed your comments raised in a previous round of review and you feel that this manuscript is now acceptable for publication, you may indicate that here to bypass the “Comments to the Author” section, enter your conflict of interest statement in the “Confidential to Editor” section, and submit your "Accept" recommendation.

Reviewer #1: All comments have been addressed

Reviewer #2: All comments have been addressed

2. Is the manuscript technically sound, and do the data support the conclusions?

Reviewer #1: Yes

Reviewer #2: Yes

3. Has the statistical analysis been performed appropriately and rigorously? 

Reviewer #1: N/A

Reviewer #2: N/A

4. Have the authors made all data underlying the findings in their manuscript fully available?

Reviewer #1: Yes

Reviewer #2: Yes

5. Is the manuscript presented in an intelligible fashion and written in standard English?

Reviewer #1: Yes

Reviewer #2: Yes

6. Review Comments to the Author

Reviewer #1: The recommended changes have been made. A small suggestion to the author

What do you think, do both the tables look more appealing in Landscape or portrait orientation?

Reviewer #2: This version is technically a thick scientific piece in terms of methodology. I am sure this paper has a high degree of novelty. This piece will help policymakers adopt evidence-based policies for human resources and health in Morocco.

7. PLOS authors have the option to publish the peer review history of their article (what does this mean?). If published, this will include your full peer review and any attached files.

Reviewer #1: **Yes: **Rajeev Mishra

Reviewer #2: **Yes: **Sanjeev Kumar
